# First Detection of Paralytic Shellfish Toxins from *Alexandrium pacificum* above the Regulatory Limit in Blue Mussels (*Mytilus galloprovincialis*) in New South Wales, Australia

**DOI:** 10.3390/microorganisms8060905

**Published:** 2020-06-16

**Authors:** Abanti Barua, Penelope A. Ajani, Rendy Ruvindy, Hazel Farrell, Anthony Zammit, Steve Brett, David Hill, Chowdhury Sarowar, Mona Hoppenrath, Shauna A. Murray

**Affiliations:** 1Climate Change Cluster (C3), University of Technology Sydney, Sydney 2007, Australia; Penelope.Ajani@uts.edu.au (P.A.A.); Rendy.Ruvindy@uts.edu.au (R.R.); Shauna.Murray@uts.edu.au (S.A.M.); 2Department of Microbiology, Noakhali Science and Technology University, Noakhali 3814, Bangladesh; 3NSW Food Authority, NSW Department of Primary Industries, PO Box 232, Taree 2430, Australia; hazel.farrell@dpi.nsw.gov.au (H.F.); Anthony.Zammit@dpi.nsw.gov.au (A.Z.); 4Microalgal Services, 308 Tucker Rd, Ormond 3204, Australia; microalgal@gmail.com (S.B.); algae@bigpond.net.au (D.H.); 5Sydney Institute of Marine Science, 19 Chowder Bay Road, Mosman 2088, Australia; Chowdhury.Sarowar@sims.org.au; 6Senckenberg am Meer, Deutsches Zentrum für Marine Biodiversitätsforschung (DZMB), Südstrand 44, 26382 Wilhelmshaven, Germany; mona.hoppenrath@senckenberg.de

**Keywords:** *Alexandrium pacificum*, paralytic shellfish toxins, blue mussels, *Mytilus galloprovincialis*

## Abstract

In 2016, 2017 and 2018, elevated levels of the species *Alexandrium pacificum* were detected within a blue mussel (*Mytilus galloprovincialis)* aquaculture area at Twofold Bay on the south coast of New South Wales, Australia. In 2016, the bloom persisted for at least eight weeks and maximum cell concentrations of 89,000 cells L^−1^ of *A. pacificum* were reported. The identity of *A. pacificum* was confirmed using molecular genetic tools (qPCR and amplicon sequencing) and complemented by light and scanning electron microscopy of cultured strains. Maximum reported concentrations of paralytic shellfish toxins (PSTs) in mussel tissue was 7.2 mg/kg PST STX equivalent. Elevated cell concentrations of *A. pacificum* were reported along the adjacent coastal shelf areas, and positive PST results were reported from nearby oyster producing estuaries during 2016. This is the first record of PSTs above the regulatory limit (0.8 mg/kg) in commercial aquaculture in New South Wales since the establishment of routine biotoxin monitoring in 2005. The intensity and duration of the 2016 *A. pacificum* bloom were unusual given the relatively low abundances of *A. pacificum* in estuarine and coastal waters of the region found in the prior 10 years.

## 1. Introduction

One of the most common and pervasive toxins produced by microalgae is saxitoxin (STX) and its analogs, also known as paralytic shellfish toxins (PSTs), which cause paralytic shellfish poisoning (PSP), a potentially fatal human illness [1,2]. PSP associated symptoms include tingling and numbness in the lips, tongue, fingers and toes, muscular weakness and breathing difficulty [1,3]. Death can occur due to respiratory failure in extreme cases [3]. These neurotoxins are produced by species of the genus *Alexandrium* Halim, and a single species of *Gymnodinium* Stein and one of *Pyrodinium* Plate in temperate coastal waters [1,4,5,6]. PSTs are highly potent, such that only 100 to 200 cells L^−1^ in marine waters are sufficient to trigger uptake into shellfish above regulatory limits [7,8].

Species of *Alexandrium* are widely distributed and can form blooms in subarctic, temperate, tropical and subtropical regions [9]. *Alexandrium* species are distinguished from one another by morphological features such as cell size, differences in shape and ornamentation of the thecal plates, the presence or absence of a ventral pore and their chain forming capacity [10]. Some morphological characteristics, such as the presence of a ventral pore and the position of the anterior attachment pore are now considered homoplastic [11], whereas other morphological characters such as the shape of the posterior sulcal plates, are considered to be very consistent and of taxonomic value [10]. About 33 species of *Alexandrium* have been recorded worldwide, of which around 10 species can potentially produce PSTs: *A. affine* (Inoue et Fukuyo) Balech; *A. andersonii* Balech; *A. pacificum* Litaker (*= A. catenella* Group IV ribotype); *A. australiense* Sh.Murray (*= A. tamarense* Group V ribotype); *A. minutum* Halim, *A. ostenfeldii* (Paulsen) Balech et Tangen; *A. catenella* (Whedon et Kofoid) Balech; *A. tamiyavanichii* Balech; and *A. taylori* Balech [12,13,14].

PSP was first reported in Australia in 1935, when typical PSP symptoms were observed following the consumption of wild mussels collected from Batemans Bay, New South Wales (NSW) [15]. In 1986, the first PSP outbreak in Australia was recorded in Port Philip Bay, Victoria, with *A. pacificum* (as *A. catenella)* as the causative organism [16,17]. Hallegraeff et al. (1991) reported the presence of *A. pacificum* (as *A. catenella*) for the first time in a phytoplankton sample from Batemans Bay, NSW. In 1989, cysts of *Alexandrium* spp. were estimated to be in excess of 300 million cells L^−1^ in the ballast water at the port of Eden in southern NSW [18,19]. To date, 11 species of *Alexandrium* have been identified in the south-eastern waters of Australia: *A. catenella* (Group 1)*, *A. affine**, *A. pacificum* (Group IV)*, *A. diversaporum* Sh.Murray et al., *A. fraterculus* (Balech) Balech, *A. margalefii* Balech, *A. minutum**, *A. ostenfeldii**, *A. pseudogonyaulax* (Biecheler) Horiguchi *ex* Yuki et Fukuyo, and *A. australiense* (Group V)* [12,14,19,20,21,22,23,24] (species marked with * are PST producing). Between 2005 and 2012, *A. pacificum* was responsible for more than 50% of algal bloom related shellfish harvest closures in NSW; however, none of these incidents were associated with PST in shellfish tissue above the regulatory limit [19]. *Gymnodinium catenatum* Graham is also present in the NSW region; however, prior to 2019, this species was not associated with PST events in shellfish harvest areas. During April 2019, a positive PST result was associated with up to 7400 cellsL^−1^ of *G. catenatum* in a Nambucca River harvest areas (NSW FA, unpublished data).

Seafood industries make a significant contribution to the Australian economy, as they harvest about 227,000 tons of seafood annually, with a value of ~$AUD 2.2 billion [1]. The farm-gate value of commercial shellfish aquaculture in New South Wales (NSW) was estimated at $AUD 54.5 million during 2017/2018. This industry has steadily increased in value since 2014/2015 (~$AUD 41 million) [25]. The largest toxic dinoflagellate bloom event in Australia occurred in 2012, when a shipment of blue mussels (*Mytilus galloprovincialis* Lamarck, 1819) from the east coast of Tasmania was found to contain PSTs above the regulatory limit by Japanese import authorities [26]. This toxic event cost the Australian industry AUD ~$23 M in lost revenue. *Alexandrium catenella* (Group 1 genotype) was identified as the source of the toxins in this region [26]. Significant levels of PST were also recorded in scallops, clams, and rock lobsters with a resulting six-month harvest closure along 350 km of the Tasmanian coastline [26].

In 2016, 2017 and 2018, *Alexandrium pacificum* was reported in the blue mussel (*Mytilus galloprovincialis*) shellfish harvest areas in Twofold Bay, NSW, an oceanic embayment on the south coast of NSW. The elevated concentrations of these species were associated with the first occurrence of PSTs above the regulatory limit since the establishment of the NSW Shellfish Quality Assurance Program in 2005. The identity of the species, its toxicity and the environmental conditions at the time of this shellfish contamination are examined in this study.

## 2. Materials and Methods

### 2.1. Water and Shellfish Sampling

Two shellfish “harvest areas” have been designated within Twofold Bay, NSW (Figure 1A,B). Three water sample sites (designated as 1, 3 and 4, Figure 1) for phytoplankton identification and enumeration and two shellfish sample sites (one for each of the harvest zones, A and B, at aquaculture leases proximate to water sites 1 and 4, respectively, Figure 1) for biotoxin testing were established in relation to these harvest areas in Twofold Bay in accordance with the NSW Marine Biotoxin Management Plan (MBMP) and the Australian Shellfish Quality Assurance Program (ASQAP) Operation Manual [27,28]. During the harvest season, fortnightly phytoplankton and monthly biotoxin samplings were conducted at these sites [27] (Appendix A). When phytoplankton cell concentrations were found to be above the specified phytoplankton action limits (PAL), and/or a report of a positive biotoxin test result occurred, the sampling frequency was increased to weekly and the initiatives like harvest area closure is taken if the cell count of species of *Alexandrium* exceeds the limit [27] (Appendix A). For species of *Alexandrium*, a biotoxin test is triggered at 200 cells L^−1^, while a closure occurs at 500 cells L^−1^ [27].

For phytoplankton identification and enumeration, 500 mL water samples were collected from a depth of 0.5 m from each site and preserved with Lugol’s iodine solution. On 24 October 2016, 500 mL of seawater was collected from the same sites for molecular characterisation (Table 1). Vertical phytoplankton net haul (20 µm mesh) samples were also collected to support species identification, culture establishment and detailed morphological and molecular investigations.

A combination of morphological, molecular, and toxicological data was collected from *Alexandrium pacificum* cultures, as well seawater, shellfish samples, and water mass characteristics across the 2016 and 2018 sampling periods (Table 1). For 2017, only cell abundance, light microscopy and mussel biotoxin testing were carried out.

### 2.2. *Alexandrium* Isolation and Culture Maintenance

Two nonaxenic monoclonal cultures (TFB_C/18 and TFB_G/18) were established from a net haul sample collected on 2nd August 2018. Single cell isolation of a species of *Alexandrium* was performed using drawn out glass pipettes (Pasteur pipettes) and a Nikon Eclipse TS100 inverted microscope (100× magnification). Isolated cells were transferred into Falcon^®^24 well culture plates containing 1 mL of five-times diluted K medium [29] without sodium silicate. Germanium dioxide was added at a concentration of 5 µg/mL to prevent diatom growth. Well plates were kept at 18 °C under a photon flux of 60–100 μmol photon m^−2^ s ^−1^ on a 12/12 h dark/light cycle (white fluorescent tubes) and checked every alternate day. After three weeks, monoclonal cultures were transferred into 20 mL of K medium in a 70 mL gamma sterile polystyrene container with polyethylene caps (Thermo Fisher Scientific, Australia, Pty., Scoresby, Vic, Australia) and grown under the same conditions outlined above. One millilitre of culture from each strain was transferred into fresh medium every three weeks to maintain healthy growing cultures.

### 2.3. Species Identification and Enumeration

Phytoplankton samples preserved with acidic Lugol’s iodide solution were concentrated by gravity-assisted membrane filtration. At first, the original 500 mL of the sample was mixed thoroughly and 300 mL was measured into a membrane filtration apparatus. The 300 mL of the sample was concentrated to 3.0 mL using gravity-assisted 5 µm membrane filtration (100× concentration). One millilitre of the concentrated sample was placed into a Sedgewick Rafter counting chamber. A minimum of 200 µL (1/5 of the entire chamber) was counted using a Zeiss Axiolab light microscope equipped with phase contrast (maximum magnification 400×) to provide an estimate of cell numbers. The threshold from this count is 50 cells/L. For examination of thecal plates, Lugol’s fixed cells were stained with Calcofluor White and examined with epifluorescence.

Light microscopy (LM) of living and fixed cultured cells was done with a Leica DMRB (Leica Microsystems GmbH) equipped with differential interference contrast and epifluorescence optics at 400 and 640 times magnification with oil immersion objectives. Digital photos were taken using a Leica DFC420C camera (Leica Microsystems GmbH). For thecal plate visualization, Lugol-fixed cells were stained with Solophenyl Flavine 7GFE 500 (Ciba Speciality Chemicals, High Point, NC, USA) as described by Chomérat et al. (2017) [30] and examined using epifluorescence.

For scanning electron microscopy (SEM), Lugol’s fixed cells from the established cultures (isolated on 2 August 2018) were placed on a 5 μm Millipore filter (Merck Millipore Darmstadt, Darmstadt, Germany), rinsed in distilled water, and dehydrated in a series of increasing ethanol concentrations (30%, 50%, 70%, 85%, 90%, and 100%), followed by chemical drying with hexamethyldisilazane at room temperature. When completely dry, the sample was mounted on a stub and sputter coated with gold-palladium (Bal-Tec SCD 050; BAL-TEC Präparationsgerätevertrieb, Wallof, Germany). Cells were observed using a Tescan VEGA3 microscope (Elekronen-Optik-Service GmbH, Dortmund, Germany) at 10 kV.

### 2.4. Molecular Characterisation

#### 2.4.1. Amplicon Sequencing and qPCR

Seawater samples collected on 24 November 2016 (Table 1) were filtered using 5 µm nitrocellulose filter (Merck Millipore) to concentrate the phytoplankton and stored at −80 °C prior to DNA extraction. DNA was extracted from the filter using MPBio FastDNA^®^ SPIN kit for Soil according to manufacturer’s protocol. DNA quality and quantity were examined using a Nanodrop (ND-1000, Thermo Scientific, Woltham, MA, USA). 18S amplicon sequencing of the V4 region was subsequently performed on the Illumina MiSeq 2 × 250 bp (bare pair) PE (paired end) standard sequencing platform (Ramaciotti Centre for Genomics, UNSW).

qPCR analysis was carried out using assays specific to *A. catenella*, *A. pacificum*, and *A. australiense* [31]. Species-specific primers were used to identify the species present in the sample (Table 2). The qPCR reactions were performed on a StepOne Plus Real-Time PCR System (Thermofisher Scientific, Waltham, MA, USA) platform with the following cycles: 95 °C for 10 s and 35 replicates of 95 °C for 15 s and 60 °C for 30 s. Melting curve analysis was performed at the end of each cycle to confirm amplification specificity by increasing the temperature to 95 °C for 10 s, then to 65 °C for 5 s, and 95 °C with ramp temperature of 0.5° C sec^−1^. Each 20 μL reaction contained 10 μL of SybrSelect™ Mix (ThermoFisher Scientific, Waltham, Massachusetts, USA), 0.5 μM of each primer, 1 μL template DNA, and 7 μL PCR-grade water. Standard curves used to quantify the cell concentrations were developed using a two known strains CS300 (*A. pacificum*) and AT-YC-H (*A. australiense*) grown in 18 °C, 12 h/12 h light cycle, and GSe medium [32].

#### 2.4.2. Bioinformatic Analysis of Amplicon Data

Bioinformatic analyses of the V4 region of 18S rDNA data were performed using QIIME 2 (v2019.1) [33]. Six sets of raw sequence data were imported, summarized using DEMUX plugin, and filtered (where, Q = 20) for further processing. The DADA2 plugin was used for trimming, chimera detection, zOTU picking and mapping with the following parameters: –p-trim-left-f 0 –p-trunc-len-f 220 –p-trim-left-r 0 –p-trunc-len-r 220. Taxonomic classification was performed by comparing with a QIIME compatible database PR2 (Protist Ribosomal database) using QIIME 2′s feature-classifier plugin. The resulting files were exported and used for downstream analysis using R-3.6.1. The programs phyloseq [34], ggplot2 [35], ape [36], qiime2R [37], tidyverse [38] packages were used to create OTU table, taxonomic table, and bar plot to examine species taxonomic delineation and relative abundance.

#### 2.4.3. DNA Extraction, PCR Amplification and DNA Sequencing

DNA was extracted from two established *Alexandrium* cultures (TFB_C/18 and TFB_G/18) cell pellets using the FastDNA spin kit for soil (MP Biomedicals, Solon, OH, USA) and the quality and the quantity of DNA was determined using a Nanodrop (Nanodrop2000; Thermo Scientific, Scoresby, Australia). Partial sequences of the rRNA genes large subunit (LSU) and small subunit (SSU) were amplified using previously published primers: D1F [39], D3B [40], SS3 and SS5 [41]. All PCR reactions were performed in 25 µL reaction volumes containing 12.5 µL of 2X Immomix (Bioline, Sydney, Australia), 7.5 pmol of each primer, 1 µg µL^−1^ of BSA (Biolabs, Arundel, Australia), 1 µL of template DNA and 8.5 µL of PCR-grade water. The thermocycling conditions consisted of an initial denaturation step of 94 °C for 2 min, followed by 35 cycles of 94 °C for 20 s, 56 °C for 30 s, and 72 °C for 1 min, followed by a final extension step of 7 min. PCR products were checked by agarose gel electrophoresis. PCR products were purified using Zymoclean™ gel DNA recovery kit (Zymo Research, Irvine, CA, USA) and sent to Macrogen (Seoul, Korea) for sequencing.

#### 2.4.4. Sequence Analysis and Phylogenetic Reconstruction

Sequences generated in this study were aligned with *Alexandrium* spp. data obtained from the GenBank reference database. The alignment algorithm ClustalW [42] was used through the Geneious software (v 9.1.2) [43] and manual inspection. Alignments were truncated to the same length. Phylogenetic trees were calculated using both a maximum likelihood (ML) and Bayesian Inference (BI) approach via Mr. Bayes 3.2.2 [44]. The ML tree was inferred using PHYML [45] with 1000 bootstraps (BS). The GTR (general time reversible) model with estimated gamma distribution was used for both analyses. BI was used to estimate the posterior probability (PP) distribution with Markov Chain Monte Carlo (MCMC) algorithm. Four heated chains with a temperature set at 0.2 were run simultaneously for 2,000,000 generations, a subsampling frequency of 1000, and a burn in length of 1000.

### 2.5. Biotoxin Analysis

#### 2.5.1. Biotoxin Testing of Shellfish (Mussel) Samples from Twofold Bay

During 2016, 2017 and 2018, mussel samples (consisting of 12–14 individuals, minimum 100 g tissue) were taken from sampling sites (Figure 1) when *Alexandrium* cell numbers exceeded the regulatory limit of 200 cells L^−1^ (Table 1). Samples were frozen and couriered on ice to Symbio Laboratories, Sydney for the analysis of paralytic shellfish toxins (PSTs), amnesic shellfish toxins (ASTs) and diarrhetic shellfish toxins (DSTs), as per the requirements in the MBMP.

At the National Association of Testing Authorities (NATA) accredited commercial laboratory, PST analysis was by high performance liquid chromatography (HPLC) [46]. Initial screening for PSTs included the analogues STX, GTX2, 3, C1, 2, GTX5, NEO, dcNEO, and GTX1, 4. If a positive result was reported, precolumn oxidation was used to confirm concentrations of STX, GTX2, 3, C1, 2, GTX5, dcSTX, dcGTX2, 3, NEO, dcNEO, GTX1,4, C3,4. AST (domoic acid (DA)), and DSTs (OA, dinophysistoxin 1 (DTX-1), dinophysistoxin 2 (DTX-2)), and pectenotoxin 2 (PTX-2) were analysed by liquid chromatography tandem mass spectrometry (LC-MS/MS) [47,48]. The lipophilic toxins cylindrospermopsin, gymnodimine, spirolide 1, azaspiracid 1, azaspiracid 2, azaspiracid 3, and yessotoxin were also included as part of this screening [48]. Positive toxin results were reported as equivalent to ≥1.00 mg/kg DA (AST), ≥0.25 mg/kg OA equivalents (DSTs) and ≥0.10 mg/kg STX equivalents (PSTs).

In brief, about 5 g of shellfish flesh was used for the toxin analyses by Lawrence method [46]. The flesh was homogenized using 3 mL of 1% acetic acid. Then, this mixture was boiled for 20 min in the water bath. This sample was allowed to cool down and the centrifugation was performed for 10 min at 3600× *g*. The supernatant was collected and the remaining pellet was mixed with 3 mL of 1% acetic acid. This prepared sample was centrifuged to separate the supernatant. These supernatants were mixed with water to get a final volume of 10 mL. A SPE C18 cartridge was used to perform the clean-up of this mixture. Oxidation of standards, PST positive reference matrix and sample was oxidised with a matrix modifier. After periodate oxidation of this sample, screening was performed, which was followed by the confirmation analyses.

#### 2.5.2. Toxin Profile Determination of *Alexandrium* Culture

Approximately 500 mL of each *Alexandrium* culture of the two established monoclonal cultures (~1.39 × 10^6^ cells of the strain TFB_C/18 and ~1.27 × 10^6^ cells of the TFB_G/18 respectively) were centrifuged, and the resulting cell pellets were immediately frozen at −80 °C until further processing. Pellets were then transported to the Sydney Institute of Marine Science (SIMS), NSW, Australia, where they were tested for the presence of STX and 13 analogues (C1, C2, GTX2, GTX1, dcGTX2, GTX3, GTX4, dcGTX3, GTX5, GTX6, dcNEO, dcSTX, NEO) by LCMS.

Briefly, cell pellets were extracted using the method of Harwood et al. (2013). Five millilitres of 1 mM acetic was then added to each sample tube and vortexed for 90 secs. The sample tube was placed into boiling water bath for 10 min, cooled at room temperature, then placed in an ultrasonic bath for 1 min and finally centrifuged for 5 min. The supernatant was then used either with or without dilution for chromatographic separation on a Thermo Scientific™ ACCELA™ UPLC system (Appendix A).

A Thermo Scientific™ Q EXACTIVE™ high resolution mass-spectrometer equipped with an electrospray ionization source was used for the analysis. The following source parameters were used in all experiments: a capillary temperature of 263 °C, a spray voltage of 3.5 kV, an auxiliary gas heater temperature of 425 °C, a sheath gas and an auxiliary gas flow rate of 50 and 13 (arbitrary units). The mass spectrometer was operated in polarity switching mode scanning across the range of m/z 100–500. LC-MS was performed on a Thermo Scientific™ ACCELA™ UPLC system by the method published by Boundy et al. (2015) [49]. Separation was carried out by an Acquity UPLC BEH Amide 130 (150 mm × 2.1 mm i.d., 1.7 µm particle size) column. Mobile phases were A. Water: Formic acid: NH_4_OH (500:0.075:0.3) and B. Acetonitrile:Water:Formic Acid (700:300:0.1). The injected volume was 5 µL. Thermo Xcalibur software (version 3.0.63, Thermo Fisher Scientific, Inc.) was used for the data analysis. Analytical standards for all of the PST analogues were obtained from National Research Council, Canada.

### 2.6. Oceanographic and Water Mass Characteristics

Prior to, and during the 2016 bloom event, water quality and oceanographic data were collected fortnightly at six locations in the bay (Figure 1) as part of a routine monitoring program in Twofold Bay (NSW DPI Lands, 2016). Field parameters including water column depth (m), dissolved oxygen (%, mg/L), turbidity (ntu), pH (pH units), salinity (ppt), conductivity (µS/cm) and temperature (°C) were measured using a YSI 6600 V2-4 WQ sonde to a maximum depth of 50 m. Additionally, a secchi disc was used to measure water clarity at each site. Laboratory parameters including suspended solids (mg/L), turbidity (ntu), ammonia as N (mg/L), ammonium (NH4^+^) 6 (mg/L), total nitrogen as N (mg/L), total phosphorus as P (mg/L) and chlorophyll a (µg/L) were measured from water samples collected using a vertical oriented Kemmerer sampler made of polycarbonate (with silicone end caps) and dropped to depths of ~1 m & ~8 m for sites W1, W2 and Bay 1; ~1 m & ~5 m for site CB1; and ~1 m & ~16 m for site Bay2. All water samples were kept on ice and immediately transported to a commercial laboratory for analysis. Furthermore, throughout the 2016 sampling campaign, oceanographic conditions were monitored using sea surface temperature data and eastward geostrophic current velocity data for each day sampled (Available online: http://oceancurrent.imos.org.au/product.php).

## 3. Results

### 3.1. Initial Light Microscope Identification and Cell Abundances (2016–2018)

In a phytoplankton sample collected on 18 October 2016, cells were identified as *Alexandrium pacificum* and enumerated to 500 cells L^−1^ at site 1 within Twofold Bay (Figure 2). The concentration increased to 89,000 cellsL^−1^ during the next week (Figure 2). While the highest cell concentrations of *Alexandrium* were reported from phytoplankton sampling sites in Twofold Bay, evidence of the bloom was detected in other south coast harvest areas [50]. Along the coastal shelf there were elevated concentrations (1200–15,000 cells L^−1^) of “*Alexandrium pacificum*” detected up to 13 km north and 21 km south of Twofold Bay. During the bloom event, PST levels did not exceed the regulatory limit in other NSW shellfish harvest areas. Maximum cell concentrations reported were 89,000 cells L^−1^, 50 cells L^−1^, and 600 cells L^−1^ during 2016, 2017, and 2018, respectively (Figure 2).

### 3.2. Mussel Toxin Analysis (2016–2018)

Positive PST results were reported in Twofold Bay during 2016, 2017 and 2018. A mussel tissue sample collected on 18 October 2016 was positive for paralytic shellfish toxins (PSTs), with a maximum concentration of 0.78 mg/kg total PSTs. Additional samples collected 24 October 2016 showed that toxin levels had increased (maximum 5.4 mg/kg total PSTs) (Figure 2). The bloom continued and maximum concentration of PSTs reported was 7.2 mg/kg on 21 November 2016. (Figure 2) Maximum concentrations reported were 7.2, 0.22 and 0.74 mg/kg PST during 2016, 2017 and 2018, respectively (Figure 2).

### 3.3. Amplicon Sequencing and qPCR (2016)

As individual species of the former *Alexandrium tamarense* species complex can be highly morphologically similar by light microscopy [14], molecular genetic methods were used to confirm identification. The analysis of amplicon sequencing data from the 24 November 2016 samples showed that of all the classified OTUs, dinoflagellates were the most abundant phylum with an average abundance of 95.05%, followed by Metazoa (1.53%) and Ciliophora (1.50%) (Figure 3A). Other phyla made up less than 2%. At the genus level, the most commonly observed genus was *Alexandrium* with an average abundance of 94.15% (Figure 3B) (Appendix A).

Finally, the phylogenetic analysis of the *Alexandrium* obtained revealed that the sequences were placed in a strongly supported clade with sequences from multiple *Alexandrium pacificum* strains from the NCBI database (Figure 4, 66.3/0.71 ML BS/Bayesian PP).

Molecular identification using qPCR assays specific to *A. pacificum, A. catenella*, and *A. australiense* was performed for 24 November 2016. All samples were dominated by *A. pacificum*. The species *A. catenella* was absent in all three tested leases, whereas *A. australiense* was present in the water sample site 1 at a very low concentration of 1 cells L^−1^. The cell density variation across the sample sites was high, ranging from 345 cells L^−1^ in water sampling site 1, 12,148 cells L^−1^ in water sampling site 3 and up to 13,456 cells L^−1^ in the water sampling site 4 (Figure 1 and Appendix A).

### 3.4. Morphological and Phylogenetic Analysis of Cultured Strains

Detailed light microscopy (LM, Figure 5) and scanning electron microscopy (SEM, Figure 6) analysis, was conducted on the two cultured monoclonal strains isolated on 2nd August 2018. The two strains conformed to the species description of *Alexandrium pacificum* (Figure 5 and Figure 6). Cells occurred mainly as single cells, but were also observed to form chains in culture (up to 12 cells). Cells were approximately 35 ± 3.1 µm wide (*n* = 15) and 36.2 ± 3 µm long (*n* = 15), contained elongated chloroplasts and a median-dorsal, sausage-shaped nucleus (Figure 5A–D). The episome was helmet-shaped and the hyposome roughly trapezoidal with a slightly longer left posterior side (Figure 5E). The cingulum was descending about one cingulum height (Figure 5F–H). The observed plate formula was APC 4′ 6′′ 6c 8s 5′′′ 2′′′′ (Figure 5F–O and Figure 6A–K). The first apical plate (1′) was rhomboidal with longer anterior right and posterior left sides of variable relative lengths (Figure 5F–H). Plate 1′ contacted the Po and Sa plates (Figure 5F–J,N). In some cells, a ventral pore was present (Figure 5I,J,O and Figure 6B arrows). The sixth precingular plate (6′′) was not triangular, an asymmetrical trapezoid shape and about as high as wide or slightly wider (Figure 5F–I,N and Figure 6C). The second antapical plate was transversely extended (Figure 6G,I). Eight sulcal plates were documented (Figure 5L,M and Figure 6K). The posterior sulcal plate (Sp) was symmetrical, pentagonal, and longer than wide (Figure 5I,L,M and Figure 6I), reached the antapex (Figure 6G–I), and sometimes had a posterior connecting pore (Figure 6J,K arrows). Sulcal plates were moderately developed (Figure 6F–J). Thecal plates were smooth with scattered pores of two size classes and distributed pairs of pores (often a smaller and a larger pore in close contact) (Figure 6L,M). Only very few apical pore complexes (APC) could be observed and none had an anterior connecting pore.

The phylogenetic analysis of the two strains isolated on 2 August 2018, using Maximum likelihood and Bayesian Inference conducted on the SSU rRNA and LSU rRNA (D1/D3) regions showed that they clustered together with other *Alexandrium pacificum* strains with full support (Figure 7A,B, Bayesian PP = 1, ML BS = 100).

### 3.5. Toxin Profile of *Alexandrium pacificum* Strains

The results from the LC-MS for the PSTs confirmed that both of the strains (TFB_C/18 and TFB_G/18) isolated from Twofold Bay on 2 August 2018 (Table 1), TFB_C/18 and TFB_G/18, produced various PST analogues. Both of the strains produce the analogues C1, C2, GTX1, GTX4, GTX5, dcSTX and NEO in different concentrations. The strain TFB_C/18 can also produce two more analogues STX and dcNEO (Table 3).

### 3.6. Oceanographic and Water Mass Characteristics

A total of 96 water samples were collected from eight sampling times between 20 September 2016 to 17 January 2017, and water mass characteristics are summarised in Table 4. The most notable water characteristics preceding the bloom event was an increase in chlorophyll a (from a background minimum of 1 µg/L to peaking at 6 µg/L at a depth of 8 m on 4 Oct 2016), and a decrease in water column temperature during the bloom (a minimum temperature of 14.6 °C at a depth of 16 m on 1 Nov 2016 with a maximum over the study period reported as 22.7 °C). There was little difference in water column structure observed (salinity/temperature profiles) at the eight sampling times between 20 September 2016 and 17 January 2017.

On 20 September 2016, a weak south-flowing current was observed in Twofold Bay. By 4 October 2016, a moderate to strong south-flowing current was reported. On 18 October 2016, a weak north-flowing current, influenced by an East Australian Current clockwise eddy located off continental shelf directly east of Twofold Bay persisted until mid-late October, when this eddy moved well north of Twofold Bay and directed warmer waters offshore. Inshore waters at this time were characterised by unusually cooler temperatures (Figure 8).

## 4. Discussion

### 4.1. The 2016–2018 *Alexandrium* Bloom Events

Since the initiation of the NSW Shellfish Quality Assurance Program in 2005, the majority of PST-positive events associated with *Alexandrium* have been reported from the NSW estuaries: Hawkesbury River; Georges River and Wagonga Inlet, with *Alexandrium pacificum* (=Group IV, formerly reported in Australia as “*A. catenella*”) as the main causative agent of PSP toxicity [20]. In October 2016, high cell densities of this same species was detected in the coastal waters offshore of Twofold Bay, NSW, an unprecedented event for this location in south eastern Australia. The maximum cell density (89,000 cells L^−1^) was also the highest cell count of this species ever reported in NSW, Australia, and led to approximately four months of shellfish harvest closures [50]. Moreover, this bloom event was so extensive that *A. pacificum* cells were detected up to 13 km north and 21 km south along the adjacent coastline [50]. In 2017 and 2018, subsequent blooms of *A. pacificum* occurred in Twofold Bay. While significant harvest days were lost to the local shellfish industry during these blooms, no illnesses due to the consumption of shellfish in Twofold Bay during these events was reported.

Confirmed detections of algal biotoxins above the regulatory limit in NSW shellfish is relatively rare [3] and prior to 2016, the maximum reported concentration of PSTs in NSW was 0.66 mg/kg, less than the harvest closure limit of 0.8 mg/kg PST equivalent. A PST concentration of ~10 mg/kg was measured during the unprecedented Tasmanian PST event which led to a worldwide product recall in 2012 [51]. That event was the first report of an *A. catenella* bloom in Australian waters and *A. catenella* continues to be the primary source of PSTs in Tasmanian water since then [51].

To get a further insight into the drivers of the 2016 unprecedented *Alexandrium pacificum* bloom in Twofold Bay, the eukaryotic microbial community and water mass data were examined. Though eukaryotic microbial interactions during *Alexandrium* blooms have rarely been studied, the link between the microbial community structure and the progression of a bloom has been reported, with evidence suggesting the growth of *Alexandrium* sp. succeeds the depletion of diatom growth [52]. During our study, the abundance of diatoms was very low during the bloom, with 94.15% of the eukaryotic microbial community comprised of *Alexandrium* sp. (Figure 3). While the available amplicon database for protists does not support the identification of *Alexandrium* to species level, we conducted phylogenetic analyses on the resultant OTUs in order to understand the *Alexandrium* species involved, with the resulting OTUs clustering with *A. pacificum* (Figure 4). Species specific qPCR also confirmed the abundance of *Alexandrium pacificum* during the 2016 bloom event. These results concur with *A. pacificum* (within the former “*Alexandrium tamarense* complex”) as the most widespread *Alexandrium* species along the New South Wales and Victorian coastal areas [20].

It is important to understand the environmental factors that significantly contribute to the initiation and termination of harmful algal blooms (HABs). These may include elevated nutrient levels, availability of micronutrients, water stratification, and seeding from benthic cysts during favourable conditions [53]. The availability of inorganic nitrogen and inorganic phosphorus can shift the eukaryotic community composition in oceans from a dinoflagellate dominated community to one of diatoms [52]. The flow of unusually cool and nutrient rich water into Twofold Bay was observed during our study and may have contributed to the unprecedented growth of this taxa at this time. Furthermore, following the initial detection of *A. pacificum* in 2016, this organism was detected again in 2017 and 2018. The presence of *Alexandrium* cysts have already been reported in the major shipping port of Eden (Twofold Bay) [54]. As high abundances of *A. pacificum* have occurred repeatedly, it is possible that cyst beds have been established in this region, and that later blooms have occurred through reseeding of the area from an established cyst bed. High rainfall, low minimum air-temperatures and low wind speed influencing *Alexandrium catenella* (Group 1 genotype) blooms has been reported for eastern Tasmania [51]. Additional data collection would be required in follow-up studies in order to characterize these factors for *Alexandrium pacificum* blooms in New South Wales.

### 4.2. *Alexandrium pacificum* Species Identification

Species belong to the *Alexandrium tamarense* complex are cryptic and cannot be distinguished from one another using light microscopy alone [15]. Some aspects of the morphological appearance of *Alexandrium* cells from Twofold Bay initially resembled that which are commonly reported for *A. catenella* (= Group I, formerly known as *A. fundyense*), for example, they were commonly found as single cells rather than in short chains, a common feature of *A. pacificum* [50]. However, further investigation using qPCR, amplicon sequencing, and later, culture isolation and characterisation, correctly identified the causative organism as *Alexandrium pacificum* (Figure 3 and Figure 4), with *Alexandrium catenella* (=Group 1) not found to be present. Morphological characterisation of the cultured strains showed the typical *A. pacificum* morphology, such as the cell size and shape with its distinct plate structure [15]. Cells were slightly longer than wide and occurred either as single cells or in chain form, with helmet-shaped episome and roughly trapezoidal hyposome [15]. This study documented the rare formation of 12 cell chains, which is the first time this has been reported for *A. pacificum* to our knowledge. The strains usually have a smooth cell surface, ornamented with many scattered small pores [15]. In our study, we also observed different sized pores, also in close contact (pairs), which has not been reported before (Figure 6L,M). The plate formula is Po, 4′ 6′′ 6c 8s 5′′′ 2′′′′, which is same as described for other *A. pacificum* strains. The irregularly rhomboidal first apical plate contacted the Po and Sa plates (Figure 5F–J,N). Moreover, we observed the presence of a ventral pore, which has not been reported for *A. pacificum* before (Figure 4B and Figure 5I,J,O arrows). The number of sulcal plates may vary from 8 to 10 [15] and in this study, we found the presence of eight sulcal plates (Figure 5L,M and Figure 6K). A pentagonal Sp plate with same length and width is commonly found in *A. pacificum* stains [15], whereas in our study, we found the length is slightly longer than the width (Figure 5I,L,M and Figure 6I). A posterior connecting pore was also present in the Sp plates as previously reported [15]. Generally, an anterior connecting pore is found in the Po [15]. However, in our study, no anterior connecting pore was observed.

Phylogenetic analysis revealed that the isolated strains clustered in a well-supported clade with other *A. pacificum* (Group IV of the *A. tamarense* species complex) strains (Figure 7A,B). Both of the trees based on LSU and SSU sequences show those strains position with other *A. pacificum* in the same clade (Figure 7A,B).

Both clonal isolates from this study produced PSTs, with little variation found in their toxin profiles. *Alexandrium pacificum* strains produce C1, C2, GTX1, GTX4, GTX5, dcSTX, dcNEO, NEO and STX in different concentrations, which has been reported in previous studies of *A. pacificum*. Different STX analogues like B1, C1, C2, GTX 1-6, dcGTX3, dcSTX, dcNEO, NEO and STX have been reported in different studies of *A. pacificum* around the world [14,22,55,56,57,58,59,60,61]. The toxin data presented in this study are indicative only, and future work should include a more rigorous investigation into the intra- and interstrain toxin variability of *A. pacificum* blooms in south eastern Australia

## 5. Conclusions

This study reports on the first contamination of blue mussels (*Mytilus galloprovincialis*) due to an unprecedented bloom of the toxic dinoflagellate *Alexandrium pacificum* in Twofold Bay, Australia. This was the first record of PSTs above the regulatory limit in commercial aquaculture in south-eastern Australia. Unequivocal identification of this species was undertaken using molecular genetic tools complemented with light and scanning electron microscopy.

This region of the world is a global hotspot for ocean warming [62]. Changes such as increasing water temperature, rainfall, salinity, and nutrient availability may influence the frequency, duration and extent of HABs in this region into the future [53]. More studies are required to determine the specific factors which influence *A. pacificum* blooms in this region.

## Figures and Tables

**Figure 1 microorganisms-08-00905-f001:**
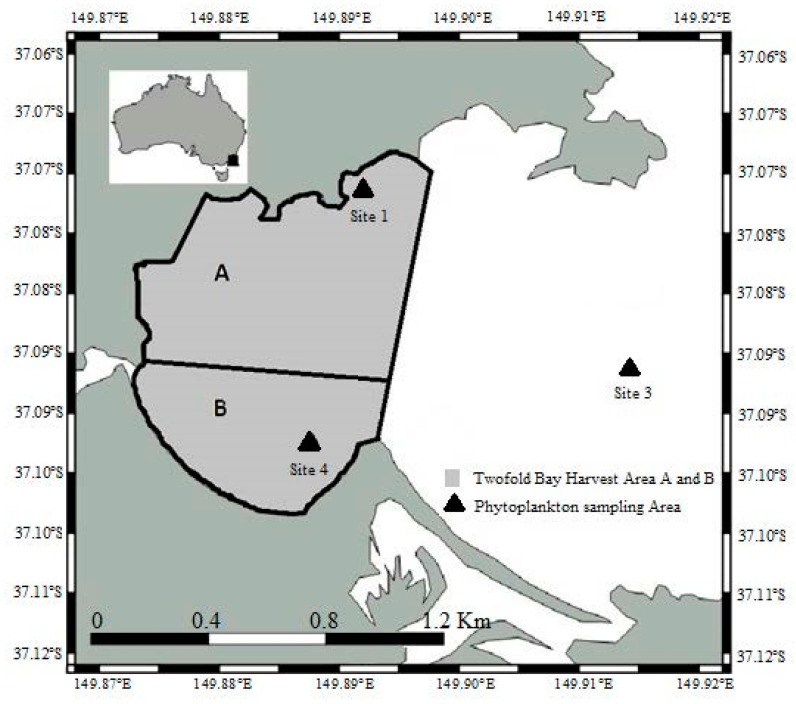
Map of Twofold Bay, New South Wales, Australia, showing sampling sites and aquaculture harvest areas A and B.

**Figure 2 microorganisms-08-00905-f002:**
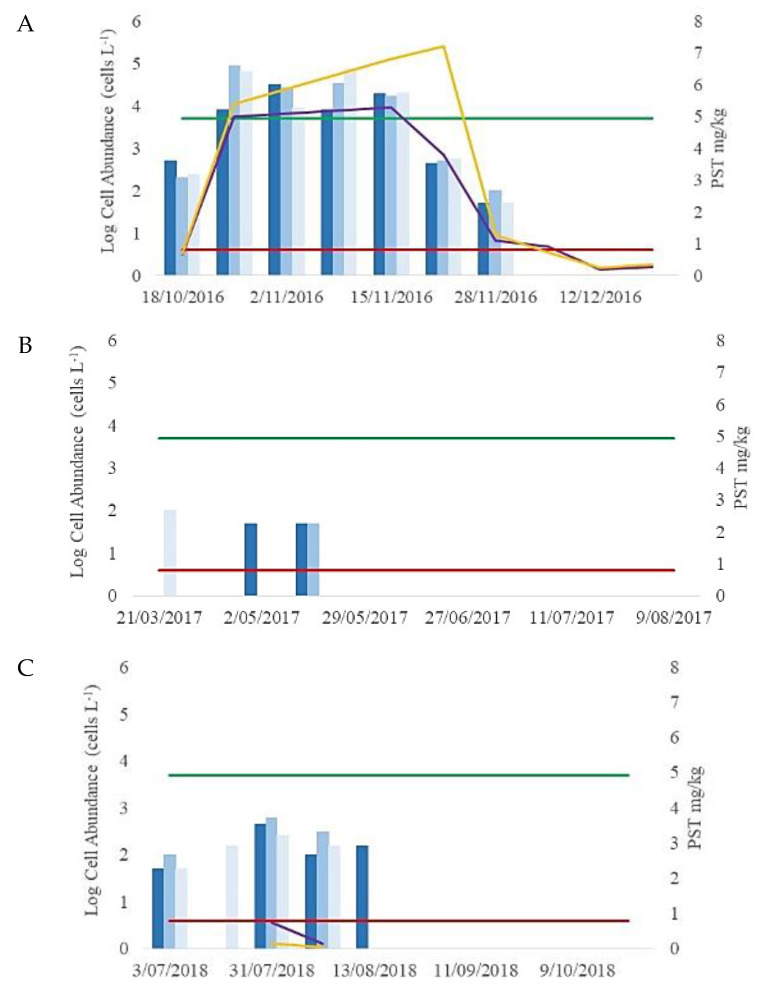
*Alexandrium pacificum* cell abundance at three sites (Site 1 dark blue; site 3 blue and site 4 light blue) across (**A**) 2016; (**B**) 2017; and (**C**) 2018 bloom events. Dark purple line is paralytic shellfish toxin (PST) concentration in mussels (mg/kg) collected from Area A; yellow line is PST concentration in mussels (mg/kg) from Area B; green line is DPI Phytoplankton Alert Level (5000 cells L^−1^) and red line is DPI Regulatory Limit for PST in shellfish (mg/kg) in shellfish.

**Figure 3 microorganisms-08-00905-f003:**
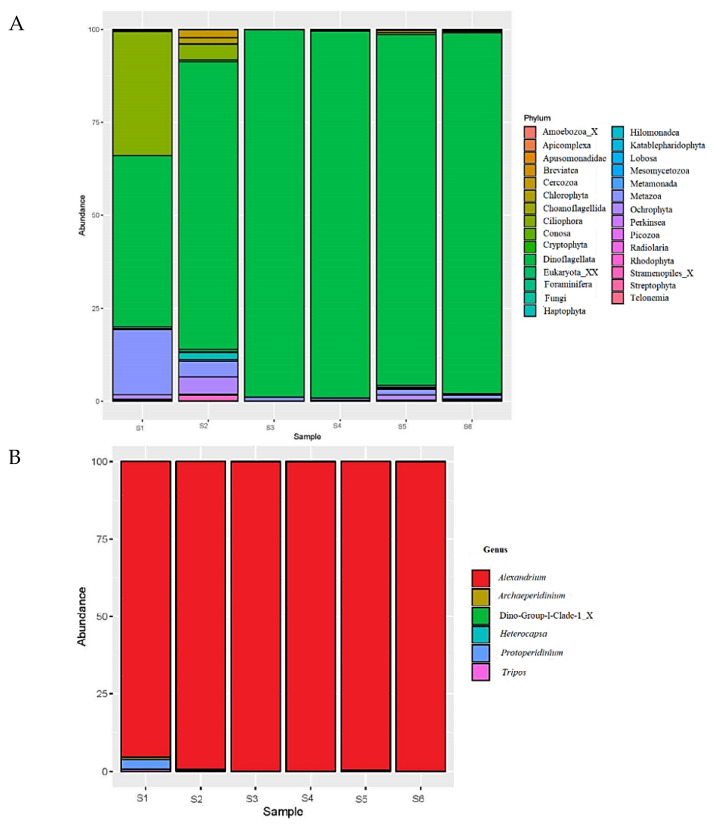
Eukaryotic community composition in samples. (**A**) Abundances of the dominant group in the phylum level. (**B**) Abundances of the dominant genus of the most dominant phylum, dinoflagellata S1 and S2 stand for sample from the sampling site 1 (Figure 1), S3 and S4 stand for sample from the sampling site 3 (Figure 1), S5 and S6 stand for sample from the sampling site 4 (Figure 1).

**Figure 4 microorganisms-08-00905-f004:**
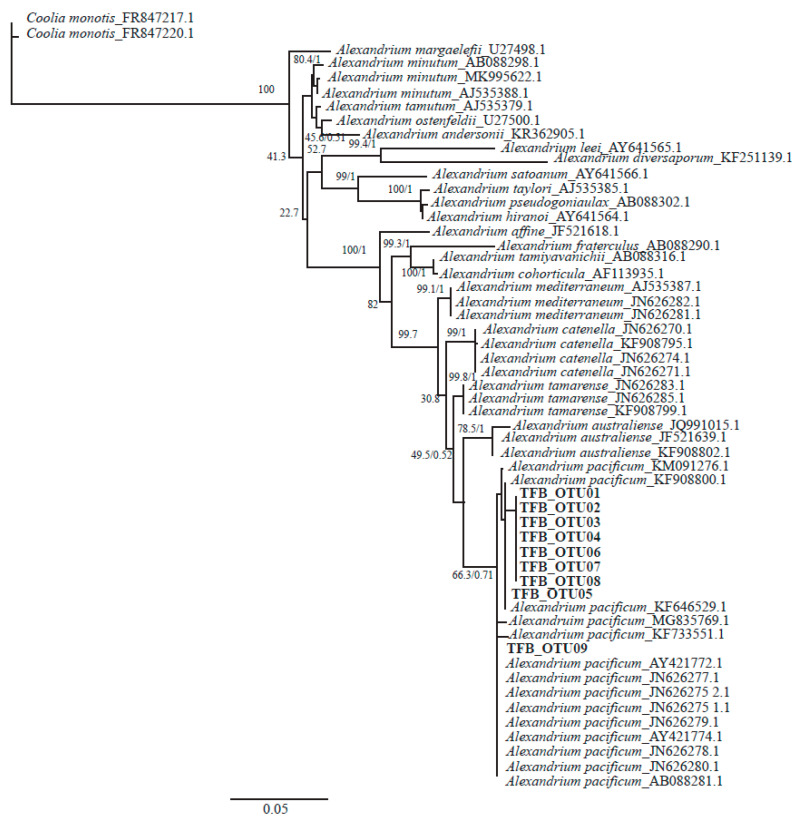
Phylogeny of the amplicon sequences of *Alexandrium*. The trees were constructed with Bayesian Inferences (MrBayes). Numbers at nodes represent posterior probabilities from Bayesian Inferences (BI) and bootstrap support values from Maximum Likelihood (ML) based on 1000 replicates.

**Figure 5 microorganisms-08-00905-f005:**
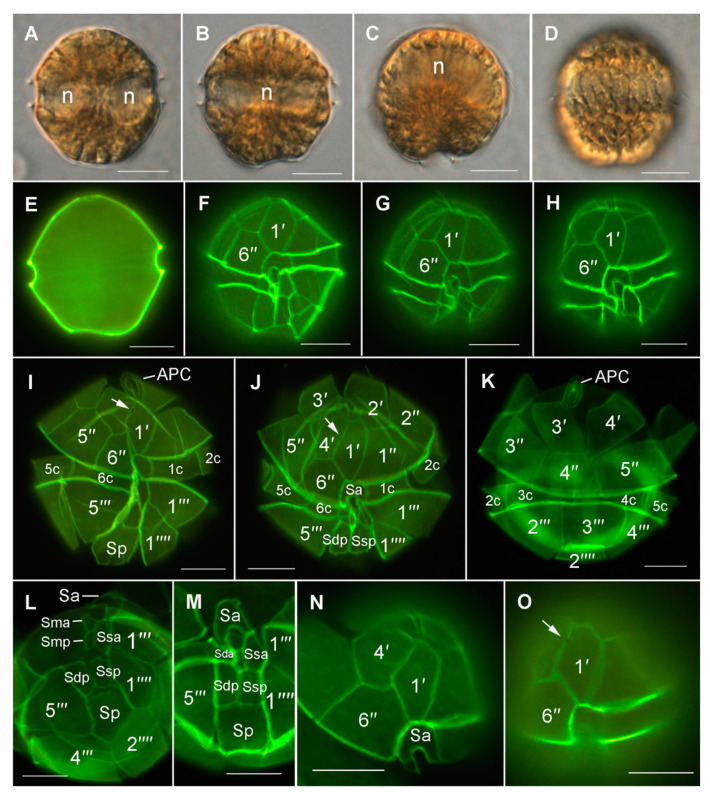
Light micrographs of *Alexandrium pacificum* using differential interference contrast (**A**–**D**) and epifluorescence microscopy (Solophenyl Flavine staining) of Lugol-fixed cultured cells (**E**–**O**). (**A**) Ventral view with focus in the cell middle, notice the two ends of the sausage-shaped nucleus (*n*). (**B**) Dorsal view with focus in the dorsal half of the cell showing the path of the nucleus (*n*). (**C**) Antapical view with focus in the cell middle, notice the sausage-shaped nucleus (*n*) dorsally. (**D**) Dorsal view with focus on the peripherally located, elongated chloroplasts. (**E**) General cell shape. (**F**–**H**) Ventral view showing the shapes of the first apical (1′) and sixth precingular (6′′) plates. Note the different degree of asymmetry in 1′. (**I**,**J**) Ventral views of squeezed thecae showing the thecal plates. (**K**) Doral view of a squeezed theca showing the thecal plates. (**L**) Ventral to antapical view showing sulcal plates. (**M**) Sulcal plates. (**N**) Ventral epithecal view showing the shapes of characteristic plates 1′, 6′′, and anterior sulcal plate (Sa). (**O**) The first apical plate (1′) with ventral pore (arrow). Scale bars = 10 µm.

**Figure 6 microorganisms-08-00905-f006:**
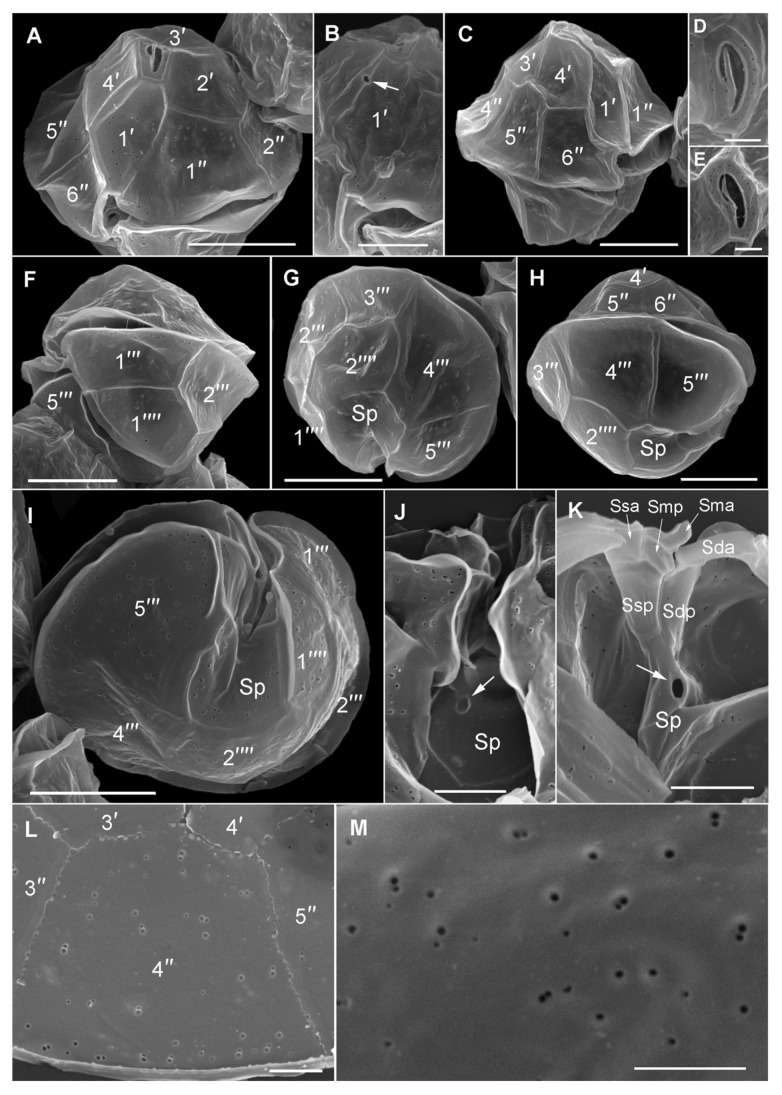
Scanning electron microscopy of *Alexandrium pacificum*. (**A**) Ventral of the epitheca. (**B**) Detail of a first apical plate (1′) with ventral pore (arrow). (**C**) Right lateral to ventral cell view. (**D**,**E**) Apical pore complex. (**F**) Left lateral view of a hypotheca. (**G**) Hypotheca in antapical view. (**H**) Right lateral cell view. (**I**) Hypotheca in antapical view showing the sulcus. (**J**) Detail of the posterior sulcal plate (Sp) with posterior connecting pore (arrow). (**K**) Inside view of the sulcal area of the hypotheca showing sulcal plate details. Note the posterior connecting pore (arrow). (**L**,**M**) Thecal pore pattern and sizes. Scale bars = 10 µm, (**B**,**J**,**K**) 5 µm, (**D**,**E**,**L**,**M**) 2 µm.

**Figure 7 microorganisms-08-00905-f007:**
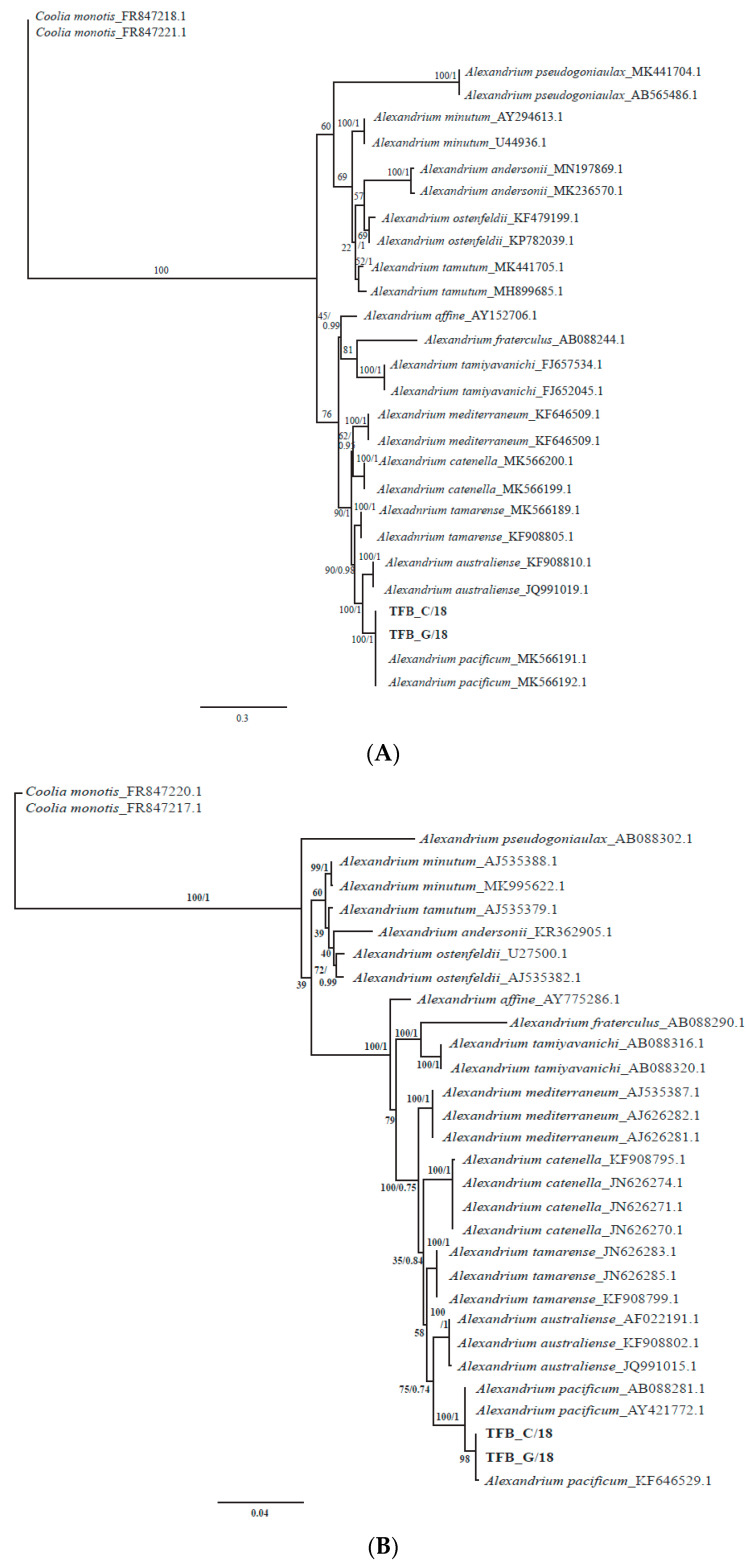
(**A**,**B**) Phylogeny of *Alexandrium*. The trees were constructed with Bayesian Inference (MrBayes). Numbers at nodes represent posterior probabilities from Bayesian Inferences (BI) and bootstrap support values from Maximum Likelihood (ML) based on 1000 replicates. (**A**) Phylogenetic tree based on large subunit (LSU) rRNA. (**B**) Phylogenetic tree based on small subunit (SSU) rRNA.

**Figure 8 microorganisms-08-00905-f008:**
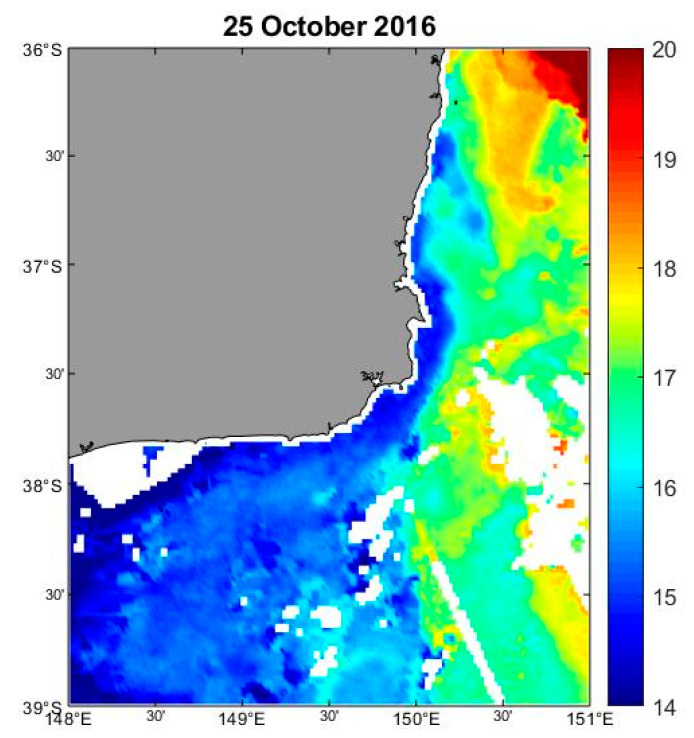
Three-day mean of sea surface temperature (SST) of the south-eastern Australian coastline on 24 October 2016 showing the warm sea-surface temperature extending from the north as red and the cold nutrient rich water extending from Bass Strait in the south as blue (Figure courtesy of Charitha Pattiaratchi, Australian National Facility for Ocean Gliders (ANFOG).

**Table 1 microorganisms-08-00905-t001:** Morphological, molecular, toxicological and water mass characterization collected for *Alexandrium pacificum* bloom identification in Twofold Bay during the 2016 and 2018 sampling periods.

Data Collected	2016	2018
**Morphological**		
Cell abundance	x	x
Strain isolation		x
Light microscopy	x	x
Scanning electron microscopy		x
**Molecular**		
Amplicon Sequencing	x	
qPCR using species specific primers	x	
LSU rDNA of cultured strain		x
SSU rDNA of cultured strain		x
**Toxicological**		
Toxin profile determination by LCMS/MS		x
Mussel biotoxin test	x	x
**Oceanographic conditions**		
Water mass characteristics	x	
Physicochemical parameters	x	
Nutrients	x	

**Table 2 microorganisms-08-00905-t002:** List of qPCR assays and primers used for *Alexandrium* species identification.

Species	Primers	
	Name	Sequence
*A. catenella*	ACT-US-408-F	5′-ACT TGA TTT GCT TGG TGG GAG-3′
	ACT-US-645-R	5′-AAG TCC AAG GAA GGA AGC ATC C-3′
*A. pacificum*	ACTA-416-F	5′-TCC TCA GTG AGA TTG TAG TG-3′
	ACTA-605-R	5′-GAC AAG GAC ACA AAC AAA TAC-3′
*A. australiense*	AusTv2-F	5′-CGG TGG GTG CAA TGA TTC-3′
	AusTv2-R	5′-GCA GGA AAA TTA CCA TTC AAG T-3′
	AusTv2-P	5′-CACAGGTAATCAAATGTCCACATAGAAACTG-3′

**Table 3 microorganisms-08-00905-t003:** Toxin data of *Alexandrium pacificum.*

Target Compounds		Strain TFB_C/18			Strain TFB_G/18	
	Total Toxin (ng)	Relative % of Total Toxin	Toxin Per Cell (pg/Cell)	Total Toxin (ng)	Relative % of Total Toxin	Toxin Per Cell (pg/Cell)
C1	143.1	2.8	0.103	349.8	6.4	0.274
C2	1893.2	36.5	1.358	821.5	15.0	0.644
GTX2	ND	0.0	ND	ND	0.0	ND
GTX1	672.2	13.0	0.482	820.8	15.0	0.643
dcGTX2	ND	0.0	ND	ND	0.0	ND
GTX3	ND	0.0	ND	ND	0.0	ND
GTX4	692.1	13.4	0.496	728.2	13.3	0.571
dcGTX3	ND	0.0	ND	ND	0.0	ND
GTX5	1110.4	21.4	0.796	2132.6	39.1	1.671
GTX6	ND	0.0	ND	ND	0.0	ND
STX	83.9	2.2	0.0602	ND	0.0	ND
dcNEO	112.7	3.0	0.0808	ND	4.3	ND
dcSTX	157.2	6.1	0.113	235.8	6.8	0.185
NEO	816.3	1.6	0.585	370.4	0.0	0.29

ND = not detected; Limit of detection for all of the targeted compounds were considered to be 0.1 ng/mL. The limit of quantification is 0.3–0.5 ng/mL.

**Table 4 microorganisms-08-00905-t004:** A summary of water quality measurements taken within Twofold Bay throughout the 2016 bloom sampling campaign. Data from each site has been combined and summarised to provide a general description of the water mass characteristics within the bay (*LOR = Limit of reporting).

		LOR	Min	Max	Mean
**A.**	Field Parameters				
	**DO** (%)	0.1	85.9	112.6	100.9
	**DO** (mg/L)	0.01	7.0	9.0	7.9
	**Turbidity** (ntu)	0.1	0.0	7.4	1.0
	**Secchi Depth** (m)	0.1	1.4	16.0	4.8
	**pH**	0.01	8.0	8.2	8.1
	**Salinity** (ppt)	-	34.9	35.9	35.5
	**Conductivity** (µS/cm)	-	52,973.0	54,203.0	53,675.9
	**Temperature** (°C)	0.01	14.6	22.7	16.7
**B.**	Laboratory parameters				
	**Suspended Solid** (mg/L)	1	1.0	40.0	6.4
	**Turbidity** (ntu)	0.1	0.2	3.2	0.7
	**Ammonia** (mg/L)	0.02	0.020	0.1	0.0
	**Ammonium** (mg/L)	0.02	0.019	0.1	0.0
	**Total Nitrogen** (mg/L)	0.05	0.1	0.6	0.2
	**Total Phosphorus** (mg/L)	0.005	0.005	0.015	0.0
	**Chl a** (µg/L)	1.0	1.0	6.0	1.9

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
