# Peer review of "First Detection of Paralytic Shellfish Toxins from Alexandrium pacificum above the Regulatory Limit in Blue Mussels (Mytilus galloprovincialis) in New South Wales, Australia"

_microorganisms, 2020, doi:10.3390/microorganisms8060905_

Round 1
Reviewer 1 Report
The manuscript can be accepted
Author Response
Reviewer #1 Comment
The manuscript can be accepted.

Reviewer 2 Report
The manuscript by Barua et al., describes PST production by Alexandrium pacificum above the regulatory limit for the first time. The manuscript describes the major organism responsible for the bloom event, toxicity as well as some environmental parameters. In general the manuscript is well written and should be of interests to scientists within the field. Below are some general comments that need to be before the manuscript can be accepted for publication.
I am not sure how reference 1 is a good citation to be used for the PSTs… there are a number of newer, peer reviewed and easily accessible references specific to the PSTs that could (and should) be used here…
The authors continually refer to the “regulatory limit” throughout the introduction. It would be helpful to mention what the regulatory limit is.
While the authors have specifically focused on dinoflagelates as the producers of PSTs and exclude the mention of cyanobacteria, I think some careful thought has to given to the language used to ensure accuracy and reduce confusion. For example Line 81. The largest toxic dinoflagellate bloom maybe have been in 2012, but I believe (sorry if I am mistaken here…) the largest toxic bloom in Australia would belong to Anabaena circinalis in the early 1990s. More specific language could help.
Line 41: PSTs are a family of toxins, not a single toxin, please change to plural.
Sequencing results should be submitted to the NCBI database and an accession number added to the manuscript
Were toxin profile measurements of the two Alexandrium isolates performed in triplicate? The authors need to perform these in triplicate and add statistical measurements to their quantification data.
All figures need to be significantly enhanced in their resolution. All figures are currently very poor quality.
Author Response
Reviewer #2 Comments
-I am not sure how reference 1 is a good citation to be used for the PSTs… there are a number of newer, peer reviewed and easily accessible references specific to the PSTs that could (and should) be used here.
One recent publication Ajani et al. 2017 has been citated to replace the previous reference 1.
-The authors continually refer to the “regulatory limit” throughout the introduction. It would be helpful to mention what the regulatory limit is.
Lines 29-31 now read as ‘This is the first record of PSTs above the regulatory limit (0.8 mg/kg) in commercial aquaculture in New South Wales since the establishment of routine biotoxin monitoring in 2005.
-While the authors have specifically focused on dinoflagellates as the producers of PSTs and exclude the mention of cyanobacteria, I think some careful thought has to given to the language used to ensure accuracy and reduce confusion. For example Line 81. The largest toxic dinoflagellate bloom maybe have been in 2012, but I believe (sorry if I am mistaken here…) the largest toxic bloom in Australia would belong to Anabaena circinalis in the early 1990s. More specific language could help.
Line 81 now reads as ‘The largest toxic dinoflagellate bloom event in Australia occurred in 2012, when a shipment of blue mussels (Mytilus galloprovincialis Lamarck, 1819) from the east coast of Tasmania was found to contain PSTs above the regulatory limit by Japanese import authorities’
- Line 41: PSTs are a family of toxins, not a single toxin, please change to plural.
Line 41 now reads as ‘These neurotoxins are produced by species of the genus Alexandrium Halim, and a single species of Gymnodinium Stein and one of Pyrodinium Plate in temperate coastal waters.’
-Sequencing results should be submitted to the NCBI database and an accession number added to the manuscript.
The sequences were submitted to the NCBI database and the accession numbers have now been added to Figure 8. The submitted data will be published on the NCBI website on 18th May.
- Were toxin profile measurements of the two Alexandrium isolates performed in triplicate? The authors need to perform these in triplicate and add statistical measurements to their quantification data.
The reviewer has raised the issue of the potential variability of PST amounts from Alexandrium isolates, and indicated that triplicate samples could be extracted to determine with greater certainty the amount of PST toxin in isolates. We agree with the reviewer that assessing variability is very important. At the same time, determining the exact amounts of toxin produced by these strains was not the focus of this study. Nevertheless, in our toxin analysis procedure, we analysed multiple extracts in relation to controls in order to be certain regarding quantification. In our previous studies, in which we analysed replicate extracts for PST amounts using our method, we found the variability is very low (unpublished data). Therefore, we believe that the current results are representative of general toxin amounts.
- All figures need to be significantly enhanced in their resolution. All figures are currently very poor quality.
The figure resolution has been increased to 300 dpi or more for all of the figures.

Reviewer 3 Report
The work is interesting, but there is an important lack of information for some data collected throughout the years under study. If it was not possible to get all the information needed, the authors should reframe the work and not explaining about what happened during these three years. They should just give information about what they found in the period of time under study, but not as a whole study from 2016-2018.
HPLC data information is also missed. The authors only refer to Lawrence method, but should give more information about how they accomplished the HPLC analyses, as they do for LC/MS/MS.
Author Response
Reviewer #3 Comments
- The work is interesting, but there is an important lack of information for some data collected throughout the years under study. If it was not possible to get all the information needed, the authors should reframe the work and not explaining about what happened during these three years. They should just give information about what they found in the period of time under study, but not as a whole study from 2016-2018.
While we agree that the data collected is disparate across sampling years (2016-2018), the data presented was from samples available for those years. The data collected provide multiple lines of evidence to describe the first reported toxic A. pacificum bloom and toxin uptake into mussels in south eastern Australian coastal waters.
- HPLC data information is also missed. The authors only refer to Lawrence method, but should give more information about how they accomplished the HPLC analyses, as they do for LC/MS/MS.
Lines 232-234 now read as ‘Samples were frozen and couriered on ice to Symbio Laboratories, Sydney for the analysis of paralytic shellfish toxins (PSTs), amnesic shellfish toxins (ASTs) and diarrhetic shellfish toxins (DSTs), as per the requirements in the MBMP.’
Line 236 now reads as ‘At the National Association of Testing Authorities (NATA) accredited commercial laboratory, PST analysis was by high performance liquid chromatography (HPLC)’
Lines 248-256 now read as ‘In brief, about 5 g of shellfish flesh was used for the toxin analyses by Lawrence method [46]. The flesh was homogenized using 3 mL of 1% acetic acid. Then this mixture was boiled for 20 mins in the water bath. This sample was allowed to cool down and the centrifugation was performed for 10 mins at 3600 × g. The supernatant was collected and the remaining pellet was mixed with 3 mL of 1% acetic acid. This prepared sample was centrifuged to separate the supernatant. These supernatants were mixed with water to get a final volume of 10 mL. A SPE C18 cartridge was used to perform the clean-up of this mixture. Oxidation of standards, PST positive reference matrix and sample were performed with a matrix modifier. After periodate oxidation of this sample, screening was performed which was followed by the confirmation analyses.’

Round 2
Reviewer 2 Report
I believe the reviewers have adequately addressed most reviewer comments and significantly improved the manuscript.
Unfortunately, I still have a problem with the quantification of toxins and I do not believe the response by the authors addresses my concern. Indeed the biological variability is a crucial issue for quantification of any molecule. To be frank, I do not see how quantification of a single sample is not scientifically sound under any circumstances. If the samples were environmental, I could understand that not having any sample left would obviously not make replicates possible. However, samples from cultured isolates can easily be repeated. The authors mention that they in fact have analysed replicates in the past, so I am unsure why this data cannot be added... I would suggest that the authors either perform the replicates and include the data as the minimum required for the presentation of quantification data or if for some circumstance unknown to me, this can’t be done, then the authors should change the presentation of the data to the relative analogue abundance for each strain.
Author Response
Reviewer #2 Comments
-Unfortunately, I still have a problem with the quantification of toxins and I do not believe the response by the authors addresses my concern. Indeed the biological variability is a crucial issue for quantification of any molecule. To be frank, I do not see how quantification of a single sample is not scientifically sound under any circumstances. If the samples were environmental, I could understand that not having any sample left would obviously not make replicates possible. However, samples from cultured isolates can easily be repeated. The authors mention that they in fact have analysed replicates in the past, so I am unsure why this data cannot be added... I would suggest that the authors either perform the replicates and include the data as the minimum required for the presentation of quantification data or if for some circumstance unknown to me, this can’t be done, then the authors should change the presentation of the data to the relative analogue abundance for each strain.
Thank you to Reviewer 2 for their additional comment in relation to this issue. In order to conduct a toxin analysis on an Alexandrium pacificum culture, approximately 500 ml of dense, exponentially growing culture must be grown in a constant temperature incubator with a light cycle. Due to the slow growth of our two A. pacificum strains (~2 months to obtain enough biomass for toxin analysis), it would not be possible to conduct additional chemical analyses in a reasonable time frame. As suggested by the reviewer, however, we present the relative percentage of each analogue for each strain (Table 3). We have also added the following sentence to lines 553-555 in the Discussion: “The toxin data presented in this study are indicative only, and future work should include a more rigorous investigation into the intra- and inter-strain toxin variability of A. pacificum blooms in south eastern Australia.”

Reviewer 3 Report
The authors should not say that they evaluate toxins in the period from 2016 to 2018, they barely have data in 2017, therefore Table 1 should not be provided and instead real information about data in that period of time should be given.
Author Response
- The authors should not say that they evaluate toxins in the period from 2016 to 2018, they barely have data in 2017, therefore Table 1 should not be provided and instead real information about data in that period of time should be given.
Whilst we agree there is only a small subset of data available for 2017 (and hence we have now removed this sampling year from Table 1), we would argue that Table 1 is still necessary to clearly show the disparate datasets collected for the 2016 and 2018 sampling periods. We originally tried to outline these unequal datasets using text and found it very confusing for the reader.
Lines 123-127 now read: “A combination of morphological, molecular, and toxicological data was collected from Alexandrium pacificum cultures, as well seawater, shellfish samples, and water mass characteristics across the 2016 and 2018 sampling periods (Table 1). For 2017, only cell abundance, light microscopy and mussel biotoxin testing were carried out.”
Table 1 now reads “Morphological, molecular, toxicological and water mass characterization collected for Alexandrium pacificum bloom identification in Twofold Bay during the 2016 and 2018 sampling periods.

This manuscript is a resubmission of an earlier submission. The following is a list of the peer review reports and author responses from that submission.
Round 1
Reviewer 1 Report
Authors describe paralytic shellfish toxin (PST) from Alexandrium pacificumabove the regulatory limit in blue mussels in New South Wales, Australia. This is novel and makes the manuscript valuable for this field of research.
1.- Some results were obtained by LCMS/MS analysis for PST identification and quantification but do not give toxicity information. Therefore authors must specify throughout the entire manuscript that they determine the toxin profile of the samples unless they perform toxicity test and provide the data.
Line 244. The subtitle should be changed.
Page 4. Table line 13 must be modified.
Line 395. The subtitle should be changed.
2.- I regard important to have the methods described in sufficient detail.
Line 257. Authors should specify the routine analysis.
Page 14. Table 3 Authors must also indicate limit of quantification.
3.- Page 8. Authors should review Figure 2. What is the meaning of blue line?
4.- Minor changes:
Introduction
Line 38 write (STX) just after saxitoxin.
Material and Methods
Line 203 delete “he”
Reviewer 2 Report
Specific comments
- Pag 3- Fig 1 The map doesn’t show the shellfish sample sites 5 and 6 mentioned in the text.
- Page 8, Fig 2. The figure must be improved. It is difficult to distinguish the patterns of the columns, the title of axis and dates. I think that the line of PST concentration in mussels is not needed. There are so few samples. You can add this information as points without joining them with a line.
- Page 8, Line 301. Dark red? maybe dark blue?
- Page 9. Fig 3. It would be better to show percentages instead of abundances. Which would allow the other non-predominant Phylum (in A) or Genera (in B) to be visualized.
- Page 9. Line 320, 327, 328 This sentence is very confusing. Please rewrite.
- Check the super index format of the abundance units: cell L-1 in all the text. It is missing for example in page 10, line 338, 339.